# Temporomandibular Joint Space Changes in Skeletal Class III Malocclusion Patients with Orthognathic Surgery

Sung-Hoon Han [1], Jae Hyun Park [2,3], Hye Young Seo [4] and Jong-Moon Chae [2,5,6,*]

1   Department of Orthodontics, Seoul St. Mary's Hospital, College of Medicine,
    The Catholic University of Korea, Seoul 06591, Republic of Korea; scherazade@hanmail.net
2   Postgraduate Orthodontic Program, Arizona School of Dentistry & Oral Health, A.T. Still University,
    Mesa, AZ 85206, USA; jpark@atsu.edu
3   Graduate School of Dentistry, Kyung Hee University, Seoul 02447, Republic of Korea
4   School of Big Data and Financial Statistics, Wonkwang University College of Natural Sciences,
    Iksan 54538, Republic of Korea; seofaith@hanmail.net
5   Department of Orthodontics, School of Dentistry, University of Wonkwang, Iksan 54538, Republic of Korea
6   Wonkwang Dental Research Institute, University of Wonkwang, Iksan 54538, Republic of Korea
*   Correspondence: jongmoon@wku.ac.kr

**Abstract:** The purpose of this retrospective study was to evaluate changes in the temporomandibular joint spaces (TMJSs) in skeletal Class III adult patients with orthognathic surgery using cone-beam computed tomography (CBCT). CBCT images taken from 26 orthognathic surgery adult patients (15 females, 11 males, average $19.6 \pm 2.8$ years at pretreatment, range 15.8–26.8 years) with skeletal Class III malocclusion (ANB < 1°) were used for this study. TMJSs (AS, anterior space; SS, superior space; PS, posterior space; MS, medial space; CS, central space; LS, lateral space) were measured at each stage of treatment (T0, pretreatment; T1, presurgery; T2, postsurgery; T3, posttreatment, and T4, retention) and were compared according to gender, side, vertical skeletal pattern, number of surgery sites, and amount of mandibular setback. At T0, TMJSs were significantly greater in SS than in AS and PS. The ratio of AS to SS to PS was 1.0 to 1.5 to 1.1. TMJSs were significantly greater in MS and CS than in LS. The ratio of MS to CS to LS was 1.0 to 1.0 to 0.8. All TMJSs in males were significantly greater than in females except in PS. TMJSs on the left side were significantly greater than on the right side only in PS. TMJSs were not significantly different depending on the SN-MP, number of surgery sites, and amount of setback. From T0 to T4, there were no significant changes in TMJSs or their ratios according to gender, side, sella to nasion (SN), mandibular plane (MP), number of surgery sites, and amount of setback. Exceptionally, at T4, SS and CS were significantly greater in the small amount of setback group than in the large amount of setback group. There were no statistical changes in TMJSs throughout all stages when skeletal Class III patients were treated with surgery.

**Keywords:** Class III malocclusion; cone-beam computed tomography; orthognathic surgery; temporomandibular joint space

## 1. Introduction

Temporomandibular joint spaces (TMJSs) have been investigated to determine the condyle–fossa relationships. The mean ratios of AS (anterior space) to SS (superior space) to PS (posterior space) in the sagittal view and LS (lateral space) to CS (central space) to MS (medial space) were evaluated. SS was the greatest and AS was the least in the sagittal view. CS was the greatest and LS was the least in the coronal view [1–3].

TMJSs were not significantly different according to skeletal patterns in our previous study [3], while the other studies showed that the condyles were positioned more anteriorly [4], posteriorly [5], and superiorly [6] in the patients with a high angle skeletal pattern than in others, and Class III patients showed more anterior position of the condyles than

other groups [7]. However, the condyle position in the TMJ did not correlate with the temporomandibular disorder (TMD) [8].

In skeletal Class III malocclusions, the dimensions of the anteroposterior glenoid fossa and AS were smaller than normal, which suggests that the mandible displaced anteriorly due to its relationship to the condylar protrusion, with a relative mediolateral elongation of the condyle within a relatively smaller glenoid fossa [9]. TMJSs were symmetrical in both condyles of the patients with facial Class III deformity and an indication for orthognathic surgery [10]. However, the incidence of internal derangement in asymmetrical patients with dentofacial Class III deformity was associated with variations in the TMJ morphology, including the disc position on both sides [11].

Mandibular prognathism with Class III skeletal deformity can be corrected by orthognathic surgery, including a sagittal split ramus osteotomy (SSRO) and intraoral vertical ramus osteotomy (IVRO) with or without maxillary surgery. Although both surgical procedures can be biologically sound, IVRO may have more favorable effects on TMJ due to condylar adaptive remodeling and anterior–inferior displacement of the condyle after surgery [12,13], and the improved anterior disc displacement in the initial postsurgical period [14].

Several factors are related to changes in the TMJS throughout the orthodontic treatment combined with orthognathic surgery. These are anterior disc displacement [15], TMJ dysfunction [16], rigid fixation procedure [17], technical problems [18], magnitude of setback [19,20], vertical bony step [21], condylar repositioning devices [22–24], biomechanical stress on the condyle [25], and vertical and horizontal skeletal patterns [3–7]. Additionally, other factors such as gender, side, number of surgery sites, type of surgery, asymmetry, genioplasty, and overbite should be considered when planning orthognathic surgery for skeletal Class III deformities.

The authors hypothesized that orthognathic surgery does not influence the TMJS in treating skeletal Class III patients. The aim of this study was to identify the effects of orthognathic surgery on TMJS throughout the treatment of skeletal Class III deformities.

## 2. Materials and Methods

### 2.1. Sample Size Calculation

A power analysis using G*Power software ver. 3.1.9.2 (Franz Faul; Chris-tian-Albrechts-Universitat, Kiel, Germany) was performed to estimate the power of the analysis using a sample size of ANOVA. An effect size f = 0.45 and a total sample size of 51 is needed, and the estimated $\alpha$ error probability was 0.05, the $\beta$ error probability was 0.20, and the power was 0.80.

### 2.2. Subjects, Eligibility Criteria, and Cone-Beam Computed Tomography (CBCT)

The study sample consisted of CBCT images at 5 stages (T0, pretreatment; T1, presurgery; T2, postsurgery; T3, posttreatment, and/or T4, retention; at least 1 year posttreatment) of 26 adult patients (11 males, 15 females; mean age, $19.6 \pm 2.8$ years) who underwent orthognathic surgery at the Wonkwang University Daejeon Dental Hospital, in Daejeon, Korea, from July 2007 to December 2017.

All of the subjects met the following inclusion criteria: (1) skeletal Class III relationship (ANB < 1°), (2) had undergone orthognathic surgery; BSSRO with or without Lefort I osteotomy, (3) CBCT images taken at T0, T1, T2, T3, and/or T4. The exclusion criteria included subjects with a severe TMD and asymmetry, and any craniofacial syndrome.

The CBCT (PSR 9000N; Asahi Alphard Vega, Kyoto, Japan) images were taken in C-mode (scan size, 2003 179 mm; voxel size, 0.39 mm; field of view, 19.97 cm). The radiologic parameters were 80 kVp, 60 mAs, and 17 s scan time. The CBCT data were saved in digital imaging and communications in medicine (DICOM) files, and Invivo6 v6.0.3 software (Anatomage, San Jose, CA, USA) was used to analyze the DICOM data to generate quantitative measurements.

Institutional review board approval to conduct the study was granted by Wonkwang University Dental Hospital (No. W2101/001-001) in Daejeon, Korea.

### 2.3. Study Design

TMJSs were investigated by two examiners (S.-H.H. and L.-K.L.) who were blinded to the patient groups. The two examiners met to discuss and agree on the landmarks before proceeding. The constructed images were re-oriented using the Invivo6 v6.0.3 software with a Frankfort horizontal plane (constructed from the right porion and orbitale) (Figure 1). TMJSs were evaluated separately on both sides.

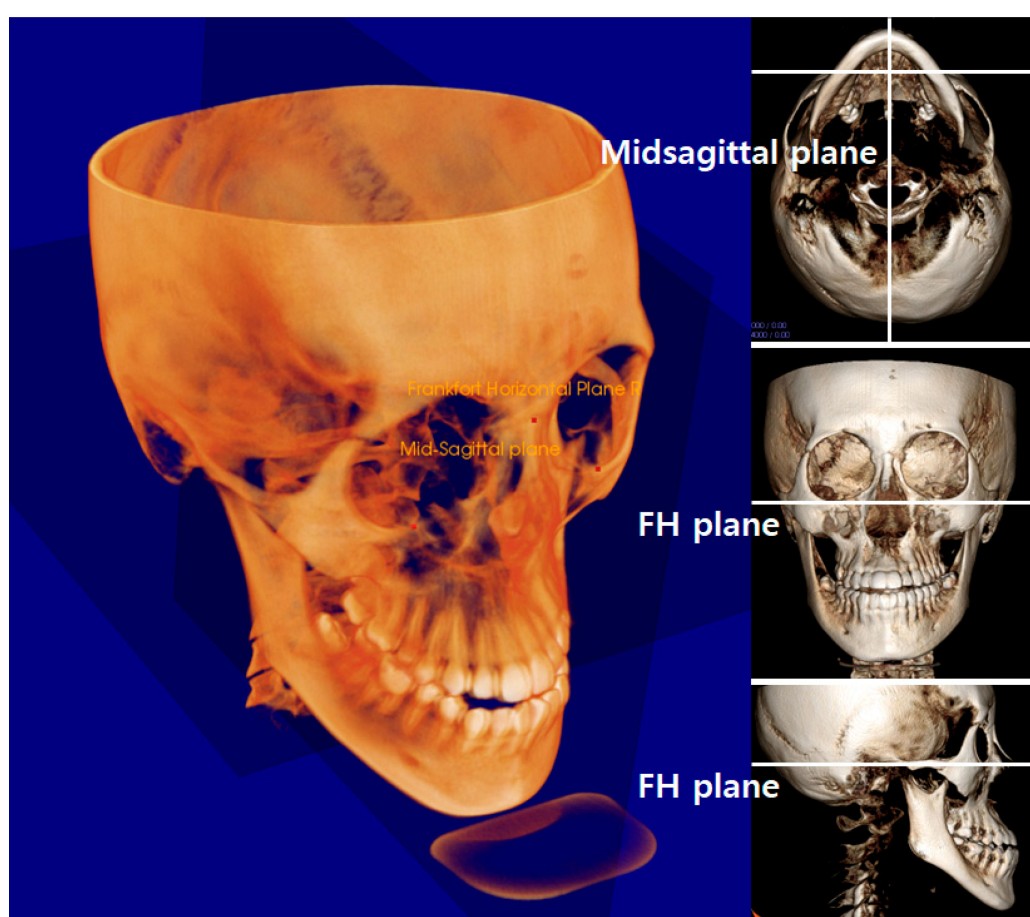

**Figure 1.** Re-orientation of cone-beam computed tomography images. Frankfurt horizontal plane: Or, orbitale; Po, porion. Midsagittal plane: Ba, basion; CG; and crista galli.

In the TMJ sectional view mode, the sagittal and coronal slices that showed the greatest anteroposterior and mediolateral dimension of the condylar head, respectively, were selected (Figure 2). The landmarks on the condyle and fossa were digitized, and TMJSs were measured using Invivo6 v6.0.3 software. TMJSs were measured as the shortest distance between 2 points at the condyle and glenoid fossa (Figure 3).

The patients were divided into five groups according to five criteria: (1) sex (female, male), (2) side (left, right), (3) vertical skeletal pattern [sella, nasion to mandibular plane (SN-MP); low $\leq 35°$ and $35° <$ high], (4) number of surgery sites (1-jaw or 2-jaw surgery), and (5) amount of mandibular setback ($\leq 6.5$ mm and $>6.5$ mm). TMJSs were measured according to the stages of treatment (T0, T1, T2, T3, and T4) and were compared according to the variables.

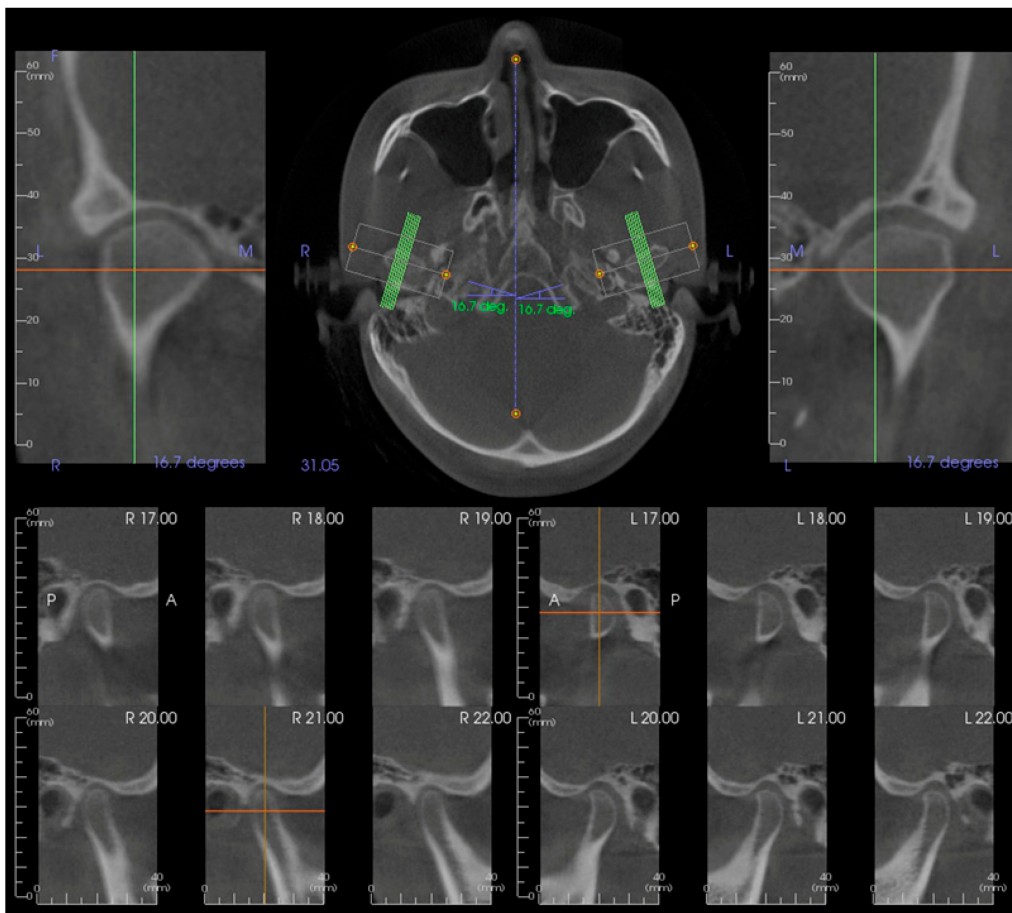

**Figure 2.** Sectional view mode of the temporomandibular joint with Invivo6 software.

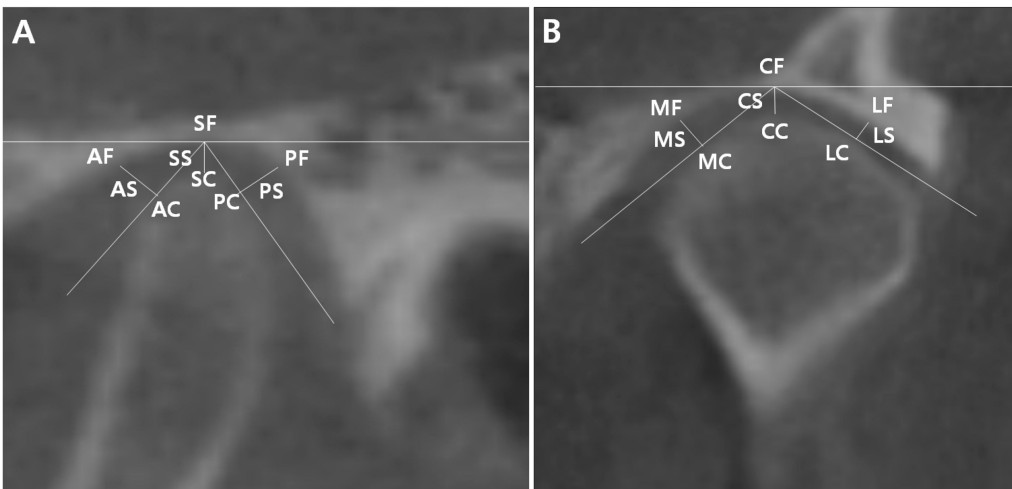

**Figure 3.** Landmarks and measurements. (**A**) Sagittal view with the greatest condylar head. AF, anterior fossa; AC, anterior condyle; AS, anterior space; SF, superior fossa; SC, superior condyle; SS, superior space; PF, posterior fossa; PC, posterior condyle; PS, posterior space. (**B**) Coronal view with the greatest condylar head. MF, medial fossa; MC, medial condyle; MS, medial space; CF, central fossa; CC, central condyle; CS, central space; LF, lateral fossa; LC, lateral condyle; LS, lateral space.

### 2.4. Statistical Analysis

An independent *t*-test and an analysis of variance were conducted to compare the TMJSs in the pretreatment coronal and sagittal CBCT images according to gender, side, SN-

MP, number of surgery sites, and amount of setback. An independent *t*-test and analysis of variance were performed to compare the TMJS changes throughout the stages of treatment (T0–T4). An independent sample *t*-test or analysis of variance was performed to compare the TMJS changes throughout the stages of treatment (T0–T4) according to the gender, side, SN-MP, number of surgery sites, and amount of setback. In the case of ANOVA, if the equal variance was not satisfied, the result of the Welch statistic has also been presented. All measurements were taken by two trained examiners. A Cronbach's Alpha test was conducted to confirm the intra- and interobserver validity.

## 3. Results

The intra- and interclass correlation coefficient (ICC) ranged from 0.83 to 0.98, showing excellent intra-and interobserver reliability.

At T0, TMJSs were significantly greater in SS than in AS and PS. The ratio of AS to SS to PS was 1.0 to 1.5 to 1.1. TMJSs were significantly greater in MS and CS than in LS. The ratio of MS to CS to LS was 1.0 to 1.0 to 0.8. All TMJSs in males were significantly greater in females except in PS. TMJSs on the left side were significantly greater than on the right side only in PS. TMJSs were not significantly different depending on the SN-MP, number of surgery sites, and amount of setback (Table 1).

**Table 1.** Temporomandibular joint space (TMJS) according to gender, side of condyle, vertical skeletal pattern, number of surgery sites, and amount of mandibular setback at pretreatment (T0).

| | | | | | | | | | | | | | | | | | | | |
|---|---|---|---|---|---|---|---|---|---|---|---|---|---|---|---|---|---|---|---|
| | | | | | | | | Mean ± Standard Deviation and Ratio | | | | | | | | | | | |
| Variables | | Total | | Gender | | | | Side | | | SN-MP | | | Number of Surgery Sites | | | Amount of Setback | | |
| CBCT | TMJS | Total (n = 52) | Ratio | Male (n = 11) | Female (n = 15) | *p* | Right (n = 26) | Left (n = 26) | *p* | Low ≤ 35° (n = 11) | High > 35° (n = 15) | *p* | 1 jaw (n = 10) | 2 jaws (n = 16) | *p* | ≤6.5 mm (n= 12) | >6.5 mm (n = 14) | *p* |
| Sagittal view | AS (mm) | 1.43a ± 0.63 | 1.0 | 1.68a ± 0.80 | 1.25a ± 0.38 | 0.029 * | 1.28a ± 0.33 | 1.58a ± 0.80 | 0.085 | 1.55a ± 0.43 | 1.34a ± 0.43 | 0.282 | 1.36a ± 0.75 | 1.47a ± 0.54 | 0.542 | 1.44a ± 0.45 | 1.43a ± 0.75 | 0.934 |
| | SS (mm) | 2.21b ± 0.89 | 1.5 | 2.73b ± 1.00 | 1.84b ± 0.57 | 0.001 ** | 2.14b ± 0.97 | 2.29b ± 0.81 | 0.543 | 2.35b ±1.11 | 2.11b ± 0.69 | 0.350 | 2.34b ±1.24 | 2.13b ± 0.59 | 0.496 | 2.15b ± 0.59 | 2.26b ±1.09 | 0.662 |
| | PS (mm) | 1.56a ±1.56 | 1.1 | 1.63a ± 0.44 | 1.50a ± 0.46 | 0.305 | 1.34a ± 0.36 | 1.77a ± 0.44 | 0.000 *** | 1.60a ± 0.45 | 1.52a ± 0.46 | 0.549 | 1.55a ± 0.52 | 1.56a ± 0.41 | 0.896 | 1.63a ± 0.38 | 1.49a ± 0.50 | 0.274 |
| | *p* | 0.000 *** | | 0.000 *** | 0.000 *** | | 0.000 *** | 0.002 ** | | 0.003 ** | 0.000 *** (0.000 ***) | | 0.002 ** (0.018 *) | 0.000 *** | | 0.000 *** | 0.000 *** | |
| Coronal view | MS (mm) | 1.96b ± 0.79 | 1.0 | 2.24a,b ± 0.98 | 1.75b ± 0.55 | 0.044 * | 1.85a,b ± 0.76 | 2.07 ± 0.82 | 0.315 | 2.09 ± 0.95 | 1.86b ± 0.65 | 0.310 | 2.15 ± 0.98 | 1.84b ± 0.63 | 0.169 | 2.01b ± 0.59 | 1.91 ± 0.94 | 0.676 |
| | CS (mm) | 2.01b ± 0.83 | 1.0 | 2.42b ± 0.95 | 1.71b ± 0.59 | 0.002 ** | 1.92b ± 0.88 | 2.10 ± 0.79 | 0.435 | 2.14 ± 0.99 | 1.91b ± 0.70 | 0.330 | 2.28 ±1.15 | 1.84b ± 0.50 | 0.117 | 1.98b ± 0.53 | 2.03 ±1.03 | 0.816 |
| | LS (mm) | 1.50a ± 0.51 | 0.8 | 1.70a ± 0.67 | 1.35a ± 0.29 | 0.030 * | 1.41a ± 0.37 | 1.60 ± 0.61 | 0.179 | 1.59 ± 0.59 | 1.44a ± 0.45 | 0.302 | 1.61 ± 0.65 | 1.44a ± 0.40 | 0.255 | 1.47a ± 0.44 | 1.53 ± 0.58 | 0.650 |
| | *p* | 0.001 ** | | 0.024 * | 0.004 ** (0.001 **) | | 0.022 * | 0.031 * | | 0.071 | .006 ** (0.002 **) | | 0.067 | 0.003 ** (0.001 **) | | 0.001 ** | 0.087 | |

CBCT, cone-beam computed tomography; TMJS, temporomandibular joint space; AS, anterior space; SS, superior space; PS, posterior space; MS, medial space; CS, central space; LS, lateral space; SD, standard deviation; SN-MP, sella-nasion to mandibular plane; Low, SN-MP ≤ 35°; High, SN-MP >35°. The letters a < b indicate vertical differences that were analyzed by Scheffe's homogeneous subset group (*p* < 0.05). The same letters mean no statistical differences; a, b. * *p* < 0.05, ** *p* < 0.01, *** *p* < 0.001.

From T0 to T4, there were no significant changes in TMJSs or their ratios according to gender, side, SN-MP, number of surgery sites, and amount of setback (Tables 2–7). Exceptionally, at T4, SS and CS were significantly greater in the small amount of setback group than in the large amount of setback group (Table 7).

**Table 2.** Temporomandibular joint space (TMJS) changes throughout the treatment stages (T0–T4).

| | | | | | | | | | | | | | |
|---|---|---|---|---|---|---|---|---|---|---|---|---|---|
| Variables | | Mean ± Standard Deviation and Ratio | | | | | | | | | | | |
| CBCT | TMJS | T0 (n = 52) | Ratio | T1 (n = 52) | Ratio | T2 (n = 52) | Ratio | T3 (n = 52) | Ratio | T4 (n = 14) | Ratio | *p* |
| Sagittal view | AS (mm) | 1.43a ± 0.63 | 1.0 | 1.57a ± 0.73 | 1.0 | 1.78a ± 1.01 | 1.0 | 1.60a ± 0.75 | 1.0 | 1.64a ± 0.51 | 1.0 | 0.271 |
| | SS (mm) | 2.21b ± 0.89 | 1.5 | 2.27b ± 0.79 | 1.4 | 2.30b ± 0.89 | 1.3 | 2.31b ± 0.80 | 1.4 | 2.22b ± 0.57 | 1.4 | 0.976 |
| | PS (mm) | 1.56a ± 1.56 | 1.1 | 1.53a ± 0.48 | 1.0 | 1.52a ± 0.45 | 0.9 | 1.53a ± 0.53 | 1.0 | 1.57a ± 0.33 | 1.0 | 0.993 |
| | *p* | 0.000 *** | | 0.000 *** | | 0.000 *** | | 0.000 *** | | 0.002 ** | | |

**Table 2.** *Cont.*

| Variables | | Mean ± Standard Deviation and Ratio | | | | | | | | | | |
| --- | --- | --- | --- | --- | --- | --- | --- | --- | --- | --- | --- | --- |
| CBCT | TMJS | T0 (n = 52) | Ratio | T1 (n = 52) | Ratio | T2 (n = 52) | Ratio | T3 (n = 52) | Ratio | T4 (n = 14) | Ratio | *p* |
| Coronal view | MS (mm) | 1.96b ± 0.79 | 1.0 | 1.88b ± 0.75 | 1.0 | 2.30b ± 1.32 | 1.0 | 2.08b ± 0.83 | 1.0 | 2.09a,b ± 0.54 | 1.0 | 0.195 |
| | CS (mm) | 2.01b ± 0.83 | 1.0 | 2.04b ± 0.68 | 1.1 | 2.07b ± 0.65 | 0.9 | 2.01b ± 0.70 | 1.0 | 2.25b ± 0.57 | 1.1 | 0.824 |
| | LS (mm) | 1.50a ± 0.51 | 0.8 | 1.49a ± 0.48 | 0.8 | 1.54a ± 0.62 | 0.7 | 1.55a ± 0.59 | 0.7 | 1.76a ± 0.41 | 0.8 | 0.583 |
| | *p* | 0.001 ** | | 0.000 *** | | 0.000 *** | | 0.000 *** | | 0.045 * | | |

CBCT, cone-beam computed tomography; TMJS, temporomandibular joint space; AS, anterior space; SS, superior space; PS, posterior space; MS, medial space; CS, central space; LS, lateral space; SD, standard deviation; T0, pretreatment; T1, presurgery; T2, postsurgery; T3, posttreatment; T4, retention. The letters a < b indicate vertical differences that were analyzed by Scheffe's homogeneous subset group ($p < 0.05$). Same letters mean no statistical differences; a, b. * $p < 0.05$, ** $p < 0.01$, *** $p < 0.001$.

**Table 3.** Comparison of temporomandibular joint space (TMJS) changes throughout the treatment stages (T0–T4) according to gender.

| Variables | | | Mean ± Standard Deviation | | | | | |
| --- | --- | --- | --- | --- | --- | --- | --- | --- |
| CBCT | TMJS | Gender | T0 (n = 52) | T1 (n = 52) | T2 (n = 52) | T3 (n = 52) | T4 (n = 14) | *p* |
| Sagittal view | AS (mm) | Male (n = 22, T4 = 10) | 1.68 ± 0.80 | 1.85 ± 0.91 | 1.91 ± 0.85 | 1.68 ± 0.78 | 1.63 ± 0.60 | 0.784 |
| | | Female (n = 30, T4 = 4) | 1.25 ± 0.38 | 1.36 ± 0.48 | 1.68 ± 1.12 | 1.54 ± 0.74 | 1.67 ± 0.22 | 0.181 |
| | | *p* | 0.029 * | 0.029 * | 0.425 | 0.517 | 0.862 | |
| | SS (mm) | Male (n = 22, T4 = 10) | 2.73 ± 1.00 | 2.66 ± 0.84 | 2.74 ± 1.02 | 2.75 ± 0.85 | 2.22 ± 0.47 | 0.577 |
| | | Female (n = 30, T4 = 4) | 1.84 ± 0.57 | 1.98 ± 0.62 | 1.97 ± 0.61 | 1.99 ± 0.59 | 2.22 ± 0.87 | 0.718 |
| | | *p* | 0.001 ** | 0.002 ** | 0.001 ** | 0.000 *** | 0.996 | |
| | PS (mm) | Male (n = 22, T4 = 10) | 1.63 ± 0.44 | 1.54 ± 0.41 | 1.51 ± 0.48 | 1.60 ± 0.54 | 1.55 ± 0.34 | 0.921 |
| | | Female (n = 30, T4 = 4) | 1.50 ± 0.46 | 1.52 ± 0.53 | 1.53 ± 0.44 | 1.48 ± 0.53 | 1.64 ± 0.35 | 0.981 |
| | | *p* | 0.305 | 0.889 | 0.917 | 0.444 | 0.680 | |
| Coronal view | MS (mm) | Male (n = 22, T4 = 10) | 2.24 ± 0.98 | 2.19 ± 0.91 | 2.31 ± 0.74 | 2.31 ± 0.72 | 2.19 ± 0.56 | 0.982 |
| | | Female (n = 30, T4 = 4) | 1.75 ± 0.55 | 1.65 ± 0.52 | 2.29 ± 1.63 | 1.92 ± 0.88 | 1.85 ± 0.47 | 0.123 |
| | | *p* | 0.044 * | 0.017 * | 0.976 | 0.090 | 0.308 | |
| | CS (mm) | Male (n = 22, T4 = 10) | 2.42 ± 0.95 | 2.37 ± 0.67 | 2.37 ± 0.57 | 2.34 ± 0.71 | 2.23 ± 0.51 | 0.970 |
| | | Female (n = 30, T4 = 4) | 1.71 ± 0.59 | 1.80 ± 0.59 | 1.85 ± 0.62 | 1.77 ± 0.59 | 2.31 ± 0.79 | 0.423 |
| | | *p* | 0.002 ** | 0.002 ** | 0.003 ** | 0.003 ** | 0.812 | |
| | LS (mm) | Male (n = 22, T4 = 10) | 1.70 ± 0.67 | 1.72 ± 0.59 | 1.58 ± 0.51 | 1.65 ± 0.48 | 1.74 ± 0.47 | 0.919 |
| | | Female (n = 30, T4 = 4) | 1.35 ± 0.29 | 1.33 ± 0.31 | 1.50 ± 0.69 | 1.47 ± 0.66 | 1.82 ± 0.22 | 0.319(0.014 *) |
| | | *p* | 0.030 * | 0.008 ** | 0.639 | 0.276 | 0.745 | |

CBCT, cone-beam computed tomography; TMJS, temporomandibular joint space; AS, anterior space; SS, superior space; PS, posterior space; MS, medial space; CS, central space; LS, lateral space; SD, standard deviation; T0, pretreatment; T1, presurgery; T2, postsurgery; T3, posttreatment; T4, retention. * $p < 0.05$, ** $p < 0.01$, *** $p < 0.001$.

**Table 4.** Comparison of temporomandibular joint space (TMJS) changes throughout the treatment stages (T0–T4) according to side.

| Variables | | | Mean ± Standard Deviation | | | | | |
| --- | --- | --- | --- | --- | --- | --- | --- | --- |
| CBCT | TMJS | Side | T0 (n = 52) | T1 (n = 52) | T2 (n = 52) | T3 (n = 52) | T4 (n = 14) | *p* |
| Sagittal view | AS (mm) | Left (n = 26, T4 = 7) | 1.58 ± 0.80 | 1.63 ± 0.89 | 1.77 ± 1.06 | 1.70 ± 0.79 | 1.70 ± 0.57 | 0.949 |
| | | Right (n = 26, T4 = 7) | 1.28 ± 0.33 | 1.51 ± 0.54 | 1.78 ± 0.98 | 1.50 ± 0.71 | 1.58 ± 0.48 | 0.134 |
| | | *p* | 0.085 | 0.565 | 0.988 | 0.336 | 0.671 | |
| | SS (mm) | Left (n = 26, T4 = 7) | 2.29 ± 0.81 | 2.33 ± 0.72 | 2.31 ± 0.85 | 2.42 ± 0.87 | 2.44 ± 0.64 | 0.970 |
| | | Right (n = 26, T4 = 7) | 2.14 ± 0.97 | 2.20 ± 0.86 | 2.28 ± 0.94 | 2.20 ± 0.72 | 2.00 ± 0.42 | 0.944 |
| | | *p* | 0.543 | 0.543 | 0.925 | 0.307 | 0.157 | |
| | PS (mm) | Left (n = 26, T4 = 7) | 1.77 ± 0.44 | 1.76 ± 0.46 | 1.68 ± 0.52 | 1.73 ± 0.52 | 1.66 ± 0.39 | 0.945 |
| | | Right (n = 26, T4 = 7) | 1.34 ± 0.36 | 1.30 ± 0.38 | 1.37 ± 0.32 | 1.33 ± 0.47 | 1.48 ± 0.26 | 0.843 |
| | | *p* | 0.000 *** | 0.000 *** | 0.012 * | 0.005 ** | 0.336 | |

**Table 4.** *Cont.*

| Variables | | | Mean ± Standard Deviation | | | | | |
|---|---|---|---|---|---|---|---|---|
| CBCT | TMJS | Side | T0 (n = 52) | T1 (n = 52) | T2 (n = 52) | T3 (n = 52) | T4 (n = 14) | *p* |
| Coronal view | MS (mm) | Left (n = 26, T4 = 7) | 2.07 ± 0.82 | 1.94 ± 0.78 | 2.48 ± 1.41 | 2.22 ± 0.78 | 2.27 ± 0.55 | 0.339 |
| | | Right (n = 26, T4 = 7) | 1.85 ± 0.76 | 1.82 ± 0.73 | 2.12 ± 1.22 | 1.95 ± 0.87 | 1.91 ± 0.51 | 0.771 |
| | | *p* | 0.315 | 0.571 | 0.327 | 0.242 | 0.222 | |
| | CS (mm) | Left (n = 26, T4 = 7) | 2.10 ± 0.79 | 2.09 ± 0.63 | 2.14 ± 0.67 | 2.12 ± 0.71 | 2.49 ± 0.68 | 0.737 |
| | | Right (n = 26, T4 = 7) | 1.92 ± 0.88 | 1.99 ± 0.73 | 1.99 ± 0.62 | 1.90 ± 0.68 | 2.01 ± 0.34 | 0.982 |
| | | *p* | 0.435 | 0.598 | 0.412 | 0.271 | 0.120 | |
| | LS (mm) | Left (n = 26, T4 = 7) | 1.60 ± 0.61 | 1.61 ± 0.57 | 1.58 ± 0.72 | 1.60 ± 0.61 | 1.76 ± 0.57 | 0.978 |
| | | Right (n = 26, T4 = 7) | 1.41 ± 0.37 | 1.38 ± 0.35 | 1.49 ± 0.51 | 1.49 ± 0.58 | 1.76 ± 0.16 | 0.350 |
| | | *p* | 0.179 | 0.094 | 0.613 | 0.502 | 0.990 | |

CT, cone-beam computed tomography; TMJS, temporomandibular joint space; AS, anterior space; SS, superior space; PS, posterior space; MS, medial space; CS, central space; LS, lateral space; SD, standard deviation; T0, pretreatment; T1, presurgery; T2, postsurgery; T3, posttreatment; T4, retention. * $p < 0.05$, ** $p < 0.01$, *** $p < 0.001$.

**Table 5.** Comparison of temporomandibular joint space (TMJS) changes throughout the treatment stages (T0–T4) according to SN-MP.

| Variables | | | Mean ± Standard Deviation | | | | | |
|---|---|---|---|---|---|---|---|---|
| CBCT | TMJS | SN-MP | T0 (n = 52) | T1 (n = 52) | T2 (n = 52) | T3 (n = 52) | T4 (n = 14) | *p* |
| Coronal view | AS (mm) | Low 35° (n = 22, T4 = 8) | 1.55 ± 0.81 | 1.69 ± 0.96 | 2.03 ± 1.41 | 1.78 ± 1.03 | 1.65 ± 0.51 | 0.629 |
| | | High > 35° (n = 30, T4 = 6) | 1.34 ± 0.43 | 1.49 ± 0.50 | 1.59 ± 0.53 | 1.46 ± 0.43 | 1.62 ± 0.57 | 0.346 |
| | | *p* | 0.282 | 0.374 | 0.170 | 0.178 | 0.934 | |
| | SS (mm) | Low 35° (n = 22, T4 = 8) | 2.35 ± 1.11 | 2.30 ± 0.94 | 2.30 ± 1.06 | 2.30 ± 0.98 | 2.10 ± 0.70 | 0.983 |
| | | High > 35° (n = 30, T4 = 6) | 2.11 ± 0.69 | 2.24 ± 0.67 | 2.30 ± 0.76 | 2.32 ± 0.66 | 2.39 ± 0.29 | 0.756 |
| | | *p* | 0.350 | 0.777 | 0.998 | 0.948 | 0.357 | |
| | PS (mm) | Low 35° (n = 22, T4 = 8) | 1.60 ± 0.45 | 1.61 ± 0.49 | 1.46 ± 0.41 | 1.41 ± 0.47 | 1.51 ± 0.38 | 0.496 |
| | | High > 35° (n = 30, T4 = 6) | 1.52 ± 0.46 | 1.47 ± 0.47 | 1.57 ± 0.49 | 1.62 ± 0.56 | 1.66 ± 0.26 | 0.750 |
| | | *p* | 0.549 | 0.283 | 0.427 | 0.150 | 0.423 | |
| | MS (mm) | Low 35° (n = 22, T4 = 8) | 2.09 ± 0.95 | 1.98 ± 0.87 | 2.60 ± 1.83 | 2.30 ± 1.03 | 2.02 ± 0.53 | 0.453 |
| | | High > 35° (n = 30, T4 = 6) | 1.86 ± 0.65 | 1.80 ± 0.66 | 2.08 ± 0.71 | 1.92 ± 0.61 | 2.18 ± 0.60 | 0.431 |
| | | *p* | 0.310 | 0.407 | 0.218 | 0.107 | 0.623 | |
| | CS (mm) | Low 35° (n = 22, T4 = 8) | 2.14 ± 0.99 | 2.18 ± 0.76 | 2.08 ± 0.67 | 2.07 ± 0.81 | 2.12 ± 0.66 | 0.992 |
| | | High > 35° (n = 30, T4 = 6) | 1.91 ± 0.70 | 1.94 ± 0.61 | 2.06 ± 0.64 | 1.96 ± 0.61 | 2.42 ± 0.43 | 0.433 |
| | | *p* | 0.330 | 0.209 | 0.894 | 0.569 | 0.355 | |
| | LS (mm) | Low 35° (n = 22, T4 = 8) | 1.59 ± 0.59 | 1.56 ± 0.46 | 1.60 ± 0.78 | 1.70 ± 0.75 | 1.72 ± 0.24 | 0.934 |
| | | High > 35° (n = 30, T4 = 6) | 1.44 ± 0.45 | 1.45 ± 0.50 | 1.49 ± 0.47 | 1.44 ± 0.43 | 1.81 ± 0.58 | 0.469 |
| | | *p* | | 0.421 | 0.528 | 0.116 | 0.688 | |

CBCT, cone-beam computed tomography; TMJS, temporomandibular joint space; AS, anterior space; SS, superior space; PS, posterior space; MS, medial space; CS, central space; LS, lateral space; SD, standard deviation; SN-MP, sella-nasion to mandibular plane; T0, pretreatment; T1, presurgery; T2, postsurgery; T3, posttreatment; T4, retention. SN-MP, sella-nasion to mandibular plane; Low, SN-MP $\leq 35°$; High, SN-MP $> 35°$.

**Table 6.** Comparison of temporomandibular joint space (TMJS) changes throughout the treatment stages (T0–T4) according to the number of surgery sites.

| Variables | | | Mean ± Standard Deviation | | | | | |
|---|---|---|---|---|---|---|---|---|
| CBCT | TMJS | Surgery Sites | T0 (n = 52) | T1 (n = 52) | T2 (n = 52) | T3 (n = 52) | T4 (n = 14) | *p* |
| Sagittal view | AS (mm) | 1 jaw (n = 20, T4 = 8) | 1.36 ± 0.75 | 1.52 ± 0.96 | 1.54 ± 0.91 | 1.58 ± 0.78 | 1.68 ± 0.44 | 0.889 |
| | | 2 jaws (n = 32, T4 = 6) | 1.47 ± 0.54 | 1.60 ± 0.56 | 1.92 ± 1.06 | 1.61 ± 0.74 | 1.58 ± 0.64 | 0.186 |
| | | *p* | 0.542 | 0.694 | 0.187 | 0.896 | 0.740 | |

**Table 6.** *Cont.*

| | Variables | | Mean ± Standard Deviation | | | | | |
|---|---|---|---|---|---|---|---|---|
| **CBCT** | **TMJS** | **Surgery Sites** | **T0 (n = 52)** | **T1 (n = 52)** | **T2 (n = 52)** | **T3 (n = 52)** | **T4 (n = 14)** | *p* |
| | | 1 jaw (n = 20, T4 = 8) | 2.34 ± 1.24 | 2.29 ± 1.01 | 2.41 ± 1.06 | 2.32 ± 1.04 | 2.26 ± 0.58 | 0.996 |
| | SS (mm) | 2 jaws (n = 32, T4 = 6) | 2.13 ± 0.59 | 2.25 ± 0.63 | 2.23 ± 0.77 | 2.30 ± 0.62 | 2.17 ± 0.61 | 0.878 |
| | | *p* | 0.496 | 0.887 | 0.479 | 0.945 | 0.762 | |
| | | 1 jaw (n = 20, T4 = 8) | 1.55 ± 0.52 | 1.49 ± 0.47 | 1.61 ± 0.55 | 1.50 ± 0.60 | 1.63 ± 0.30 | 0.927 |
| | PS (mm) | 2 jaws (n = 32, T4 = 6) | 1.56 ± 0.41 | 1.55 ± 0.49 | 1.47 ± 0.39 | 1.55 ± 0.49 | 1.50 ± 0.39 | 0.905 |
| | | *p* | 0.896 | 0.660 | 0.271 | 0.724 | 0.497 | |
| | | 1 jaw (n = 20, T4 = 8) | 2.15 ± 0.98 | 1.98 ± 0.94 | 2.17 ± 0.78 | 2.06 ± 0.83 | 2.09 ± 0.59 | 0.958 |
| | MS (mm) | 2 jaws (n = 32, T4 = 6) | 1.84 ± 0.63 | 1.82 ± 0.61 | 2.38 ± 1.57 | 2.10 ± 0.84 | 2.09 ± 0.54 | 0.140 |
| | | *p* | 0.169 | 0.467 | 0.575 | 0.855 | 0.995 | |
| Coronal view | | 1 jaw (n = 20, T4 = 8) | 2.28 ± 1.15 | 2.12 ± 0.86 | 2.12 ± 0.72 | 2.07 ± 0.91 | 2.25 ± 0.53 | 0.950 |
| | CS (mm) | 2 jaws (n = 32, T4 = 6) | 1.84 ± 0.50 | 1.99 ± 0.55 | 2.04 ± 0.60 | 1.97 ± 0.54 | 2.26 ± 0.67 | 0.424 |
| | | *p* | 0.117 | 0.525 | 0.636 | 0.640 | 0.978 | |
| | | 1 jaw (n = 20, T4 = 8) | 1.61 ± 0.65 | 1.53 ± 0.58 | 1.61 ± 0.71 | 1.53 ± 0.59 | 1.90 ± 0.45 | 0.664 |
| | LS (mm) | 2 jaws (n = 32, T4 = 6) | 1.44 ± 0.40 | 1.47 ± 0.42 | 1.49 ± 0.56 | 1.56 ± 0.60 | 1.57 ± 0.26 | 0.881 |
| | | *p* | 0.255 | 0.647 | 0.527 | 0.854 | 0.129 | |

CBCT, cone-beam computed tomography; TMJS, temporomandibular joint space; AS, anterior space; SS, superior space; PS, posterior space; MS, medial space; CS, central space; LS, lateral space; SD, standard deviation; T0, pretreatment; T1, presurgery; T2, postsurgery; T3, posttreatment; T4, retention.

**Table 7.** Comparison of temporomandibular joint space (TMJS) changes throughout the treatment stages (T0–T4) according to the amount of setback.

| | Variables | | Mean ± Standard Deviation | | | | | |
|---|---|---|---|---|---|---|---|---|
| **CBCT** | **TMJS** | **Amount of Setback** | **T0 (n = 52)** | **T1 (n = 52)** | **T2 (n = 52)** | **T3 (n = 52)** | **T4 (n = 14)** | *p* |
| | | Small < 6.5 (n = 24, T4 = 4) | 1.44 ± 0.45 | 1.54 ± 0.52 | 1.93 ± 1.22 | 1.58 ± 0.82 | 1.61 ± 0.40 | 0.288(0.505) |
| | AS (mm) | Large > 6.5 (n = 28, T4 = 10) | 1.43 ± 0.75 | 1.60 ± 0.88 | 1.65 ± 0.79 | 1.61 ± 0.71 | 1.65 ± 0.57 | 0.828 |
| | | *p* | 0.934 | 0.776 | 0.320 | 0.884 | 0.889 | |
| | | Small < 6.5 (n = 24, T4 = 4) | 2.15 ± 0.59 | 2.27 ± 0.67 | 2.26 ± 0.83 | 2.25 ± 0.69 | 2.68 ± 0.63 | 0.736 |
| Sagittal view | SS (mm) | Large > 6.5 (n = 28, T4 = 10) | 2.26 ± 1.09 | 2.27 ± 0.88 | 2.33 ± 0.95 | 2.36 ± 0.89 | 2.04 ± 0.45 | 0.911 |
| | | *p* | 0.662 | 0.997 | 0.778 | 0.628 | 0.050 * | |
| | | Small < 6.5 (n = 24, T4 = 4) | 1.63 ± 0.38 | 1.62 ± 0.50 | 1.55 ± 0.30 | 1.53 ± 0.48 | 1.76 ± 0.17 | 0.782 |
| | PS (mm) | Large > 6.5 (n = 28, T4 = 10) | 1.49 ± 0.50 | 1.46 ± 0.45 | 1.50 ± 0.56 | 1.54 ± 0.58 | 1.50 ± 0.36 | 0.987 |
| | | *p* | 0.274 | 0.236 | 0.671 | 0.948 | 0.197 | |
| | | Small < 6.5 (n = 24, T4 = 4) | 2.01 ± 0.59 | 1.95 ± 0.61 | 2.47 ± 1.76 | 2.12 ± 0.95 | 1.93 ± 0.50 | 0.466 |
| | MS (mm) | Large > 6.5 (n = 28, T4 = 10) | 1.91 ± 0.94 | 1.82 ± 0.86 | 2.15 ± 0.77 | 2.05 ± 0.73 | 2.15 ± 0.57 | 0.538 |
| | | *p* | 0.676 | 0.544 | 0.386 | 0.774 | 0.513 | |
| | | Small < 6.5 (n = 24, T4 = 4) | 1.98 ± 0.53 | 2.11 ± 0.55 | 2.11 ± 0.63 | 1.96 ± 0.63 | 2.80 ± 0.68 | 0.111 |
| Coronal view | CS (mm) | Large > 6.5 (n = 28, T4 = 10) | 2.03 ± 1.03 | 1.98 ± 0.78 | 2.03 ± 0.67 | 2.05 ± 0.76 | 2.03 ± 0.36 | 0.998 |
| | | *p* | 0.816 | 0.502 | 0.643 | 0.670 | 0.015 * | |
| | | Small < 6.5 (n = 24, T4 = 4) | 1.47 ± 0.44 | 1.47 ± 0.46 | 1.65 ± 0.74 | 1.62 ± 0.69 | 1.64 ± 0.34 | 0.735 |
| | LS (mm) | Large > 6.5 (n = 28, T4 = 10) | 1.53 ± 0.58 | 1.51 ± 0.51 | 1.44 ± 0.48 | 1.49 ± 0.50 | 1.81 ± 0.43 | 0.408 |
| | | *p* | 0.650 | 0.745 | 0.215 | 0.445 | 0.493 | |

CBCT, cone-beam computed tomography; TMJS, temporomandibular joint space; AS, anterior space; SS, superior space; PS, posterior space; MS, medial space; CS, central space; LS, lateral space; SD, standard deviation; T0, pretreatment; T1, presurgery; T2, postsurgery; T3, posttreatment; T4, retention. * $p < 0.05$.

## 4. Discussion

Ikeda and Kawamura [1] investigated TMJS to assess the optimal condylar position. They suggested that the ratio of AS to SS to PS was 1.0 to 1.9 to 1.6, similar to this study and our previous [3] studies, even though this study included only skeletal Class III patients with an ANB below 1.0°. Class III patients exhibited the lowest AS, the highest SS [7], and more anterior condylar positions [26]—findings that are similar to our present and previous [3] studies. But our study showed the ratio of MS to CS to LS was 1.0 to 1.0 to 1.8, which is different from the ratios in previous studies [2,3]. Therefore, further studies are recommended to determine not only sagittal but also coronal TMJSs according to various skeletal patterns.

Pretreatment TMJSs were greater in males than in females except for PS in this study, which differs from the results in previous studies [1–3,27] that showed no difference relative to gender in the TMJS. This study reported that PS was greater on the left than the right sides, which is consistent with previous studies [3,26,27] that showed higher PS due to the greater anterior position of the left condyle. But some studies showed the opposite results [28,29], or no differences [30,31] between the two sides. Our present study reported that there was no statistical difference in pretreatment (T0) TMJSs according to SN-MP, which is consistent with our previous study [3], but some studies showed results different from those in this study according to vertical and horizontal skeletal patterns [4–7]. These variable results might be due to the inconsistent research materials and methods. Therefore, further studies with increased sample sizes should be considered to achieve more reliable results.

There are several factors contributing to changes in TMJS throughout the orthognathic surgery approach. In this study, gender, side, SN-MP, number of surgery sites, and amount of setback were not significant factors in TMJS changes during and after treatment. Kim et al. [32] proposed that condylar angulations are more stable in one-jaw surgeries than in two-jaw surgeries, but condylar displacements were not clinically significant between the one-jaw and two-jaw groups, results that are similar to those of our study. TMJSs can be changed during mandibular setback surgery by altering the position of the proximal segment, but they tend to go back to their original position after surgery. Therefore, TMJS should be maintained by overcoming the technical problems to improve postsurgical stability [17,18,21–24]. Moroi et al. [19] suggested that the magnitude of the setback was not an influencing factor affecting bite force or occlusal contact area. This seems to be consistent with our findings that show no TMJS changes relative to the amount of setback. Relapse after SSRO is one of the postoperative complications caused by the creation of gaps between proximal and distal segments, condylar malposition after intermaxillary fixation, pterygomasseteric tension, and the rotation of the proximal segment [33]. TMJS may change if relapse occurs. Therefore, TMJS should be checked throughout the orthognathic surgical treatment to evaluate and minimize relapse.

Sagittal split ramus osteotomy (SSRO) and intraoral vertical ramus osteotomy (IVRO) are orthognathic surgery procedures used mainly to correct the Class III prognathic mandible. Each of the surgical procedures has its pros and cons. The advantages of SSRO include such things as a quicker recovery of oral function due to an easier rigid fixation, but it may cause a greater change in the condylar position and a higher incidence of TMJ problems than IVRO [12–14]. Lee et al. [34] suggested that unilateral IVRO (UIVRO) with contralateral SSRO may improve the TMJ conditions in the treatment of rotational mandibular asymmetry. On the other hand, the condyle may undergo a remodeling process with a resorptive pattern following orthognathic surgery [35]. In this study, we only evaluated the changes in TMJSs in patients with SSRO. Therefore, further studies should include patients with IVRO or UIVRO to evaluate both changes in TMJS and condyle shape.

## 5. Conclusions

Within the limitations of this retrospective study, TMJSs and their changes were evaluated with CBCT in skeletal Class III adult patients throughout the orthognathic surgery approach.

- At T0, TMJSs were significantly greater in SS than in AS and PS. The ratio of AS to SS to PS was 1.0 to 1.5 to 1.1.
- TMJSs were significantly greater in MS and CS than in LS. The ratio of MS to CS to
- LS was 1.0 to 1.0 to 0.8.
- All TMJSs in males were significantly greater than in females except in PS.
- TMJSs on the left side were significantly greater than on the right side only in PS.
- TMJSs were not significantly different depending on the SN-MP, number of surgical sites, or amount of setback.
- From T0 to T4, there were no significant changes in TMJSs and their ratios according to gender, side, SN-MP, number of surgery sites, and the amount of setback.
- Exceptionally, at T4, SS and CS were significantly greater in the small setback group than in the large setback group.

Consequently, orthognathic surgery might be a safe procedure that can be used to hold the position of the condyle throughout treatment.

**Author Contributions:** Conceptualization, S.-H.H. and J.H.P.; methodology, S.-H.H.; software, S.-H.H.; validation, S.-H.H. and J.H.P.; formal analysis, H.Y.S.; investigation, S.-H.H.; resources, J.-M.C.; data curation, H.Y.S.; writing—original draft preparation, S.-H.H. and J.H.P.; writing—review and editing, J.H.P. and J.-M.C.; visualization, S.-H.H.; supervision, J.-M.C.; project administration, J.-M.C. All authors have read and agreed to the published version of the manuscript.

**Funding:** This research received no external funding.

**Institutional Review Board Statement:** This study was approved by the Institutional Review Board of Wonkwang University Daejeon Dental Hospital (W2101/001-001).

**Informed Consent Statement:** Written informed consent was obtained from the patient for publication of this short report and any accompanying images.

**Data Availability Statement:** The authors declare that the materials are available.

**Acknowledgments:** We wish to acknowledge LK Ham for his detailed investigation of the data. This paper was supported by The Korean Orthodontic Research Institute Inc.

**Conflicts of Interest:** The authors declare no conflict of interest.

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
