# Peer review of "Temporomandibular Joint Space Changes in Skeletal Class III Malocclusion Patients with Orthognathic Surgery"

_applsci, doi:10.3390/app13169241_

Round 1

Reviewer 1 Report

Thank you for the opportunity to review this paper.

Overall it's a great work, well written. I'd appreciate if the results section in the abstracts and in the conclusion will deal more with t0-t4 differences and not t0 in-group results.

Also, don't see explanation of rt-lt side difference - more proper explanation is needed for this phenomenon.

Author Response

1. Overall it's a great work, well written. I'd appreciate if the results section in the abstracts and in the conclusion will deal more with t0-t4 differences and not t0 in-group results.

-- The authors have added an additional result with T0-T4 differences in the abstract and conclusions sections according to the reviewer’s comment.

 “Exceptionally, at T4, SS and CS were significantly greater in small amounts of the setback group than in large amounts of the setback group.”

2. Also, don't see explanation of rt-lt side difference - more proper explanation is needed for this phenomenon.

-- Page 9, lines 251-254; In the discussion section, the authors have mentioned the controversial results depending on the left and right sides, referring to some references.

Reviewer 2 Report

I would like to appreciate all authors for their work in this manuscript. However, I have mentioned some comments inside the pdf file as a sticly notes. Kindly go through the comments and take necessary action.

Best of luck.

Thank you

Author Response

I would like to appreciate all authors for their work in this manuscript. However, I have mentioned some comments inside the pdf file as a sticly notes. Kindly go through the comments and take necessary action.

-- Yes, the authors have revised the manuscript according to the reviewer’s comment.

Reviewer 3 Report

Dear Authors,

This is an interesting topic about TMJ spaces after orthognathic surgery. There are also previous reports about this subject. Your results are paralel to previous reports.  More detailed and multicentered studies are also necessary about this issue to improve the clinical knowledge. 

Best regards. 

Author Response

This is an interesting topic about TMJ spaces after orthognathic surgery. There are also previous reports about this subject. Your results are paralel to previous reports. More detailed and multicentered studies are also necessary about this issue to improve the clinical knowledge.

-- Yes, I think that you are absolutely right. I hope multicentered studies with more samples at multiple clinics will be performed.